# Enhancing Microbiome Research through Genome-Scale Metabolic Modeling

Nana Y. D. Ankrah,[a] David B. Bernstein,[b] Matthew Biggs,[c] Maureen Carey,[d] Melinda Engevik,[e] Beatriz García-Jiménez,[f] Meiyappan Lakshmanan,[g,h] Alan R. Pacheco,[i] Snorre Sulheim,[j] Gregory L. Medlock[k]

[a]State University of New York at Plattsburgh, Plattsburgh, New York, USA
[b]University of California, Berkeley, California, USA
[c]Sestina Bio, Pleasanton, California, USA
[d]University of Virginia, Charlottesville, Virginia, USA
[e]Medical University of South Carolina, Charleston, South Carolina, USA
[f]Biome Makers, Inc., West Sacramento, California, USA
[g]Bioprocessing Technology Institute, Agency for Science, Technology and Research (A*STAR), Singapore
[h]Bioinformatics Institute, Agency for Science, Technology and Research (A*STAR), Singapore
[i]ETH Zurich, Zurich, Switzerland
[j]SINTEF Industry, Trondheim, Norway
[k]Vedanta Biosciences, Cambridge, Massachusetts, USA

**ABSTRACT** Construction and analysis of genome-scale metabolic models (GEMs) is a well-established systems biology approach that can be used to predict metabolic and growth phenotypes. The ability of GEMs to produce mechanistic insight into microbial ecological processes makes them appealing tools that can open a range of exciting opportunities in microbiome research. Here, we briefly outline these opportunities, present current rate-limiting challenges for the trustworthy application of GEMs to microbiome research, and suggest approaches for moving the field forward.

**KEYWORDS** microbiome, metabolic modeling, metabolism

## OPPORTUNITIES

Genome-scale metabolic models (GEMs) are mathematical representations of the chemical reactions that can be carried out by an organism. In their simplest form, they encode the quantities of reactants and products consumed or produced in chemical reactions. As such, GEMs can simulate the growth of organisms in diverse environmental contexts through approaches such as flux balance analysis (1) and have become essential tools for generating testable hypotheses about their metabolic functions. This combination of versatility and mechanistic insight allows GEMs to be readily extended to investigate the roles of individual species in complex microbial assemblages (2–4). In particular, a collection of GEMs can predict the metabolic interactions that can emerge between members of microbial communities by calculating the flow of metabolites through each organism and its environment (5). As such, GEM simulations can also predict environmental modifications that can occur as a result of metabolite secretion and utilization by community members (6). GEMs can be used with experimental or observational data (referred to as "context-specific modeling") (7) and in time- and spatially resolved simulations with single or multiple organisms (8, 9). These approaches can all be extended to multicompartment models to study host-microbe interactions by simultaneously modeling host and microbial metabolism. In particular, such frameworks have been used to predict microbiome responses to interventions that are challenging to represent experimentally (e.g., dynamic environments, invading species in sensitive ecosystems, clinical scenarios) (10, 11). GEMs also provide

Address correspondence to Gregory L. Medlock, gmedlo@gmail.com.

The authors declare no conflict of interest.

opportunities to generate insights from microbiome multiomics, i.e., metagenomics, metatranscriptomics, metaproteomics, and/or metabolomics data (12–15).

Most microbiome studies employing GEMs use them to explicitly model a microbiome (i.e., a single simulation captures metabolism of the entire microbiome). Alternatively, the physiological factors represented by a GEM (e.g., biochemical capabilities, genome annotations) or the results of simulations with individual organisms' GEMs (e.g., simulated nutrient preferences, simulated metabolite production) can be used outside the context of microbiome-wide *in silico* simulations to evaluate new hypotheses with metagenomics and metabolomics data (12, 16, 17). For example, rather than using GEMs to simulate fluxes in a community, simulations with GEMs for individual organisms can be used to predict metabolic traits for members of a community. Statistical analyses (e.g., regression, classification) can then be performed using these traits as input to predict environmental observations (e.g., nutrient abundances, plant yields, clinical outcomes) (17). These applications, as well as their use across a variety of systems and temporospatial scales, have been thoroughly reviewed (18–21).

## CHALLENGES

To ensure the reliability of GEM predictions, it is crucial to establish the appropriate context for their application to microbial communities (22). First, one must consider that the application of GEMs is generally limited to metabolic interactions, while other factors key to microbial community dynamics (e.g., gene regulation, expression, and protein localization; pH' temperature; antibiotics; and quorum sensing) are only accounted for in specific modeling extensions (8, 23–33). Second, one should select a simulation scope according to the experimental question raised, preferred simulation output, appropriate assumptions, and available data (Fig. 1).

The application of flux balance analysis (FBA) to microbial communities necessitates the study of many nonmodel organisms, requiring either reconstruction of strain-specific GEMs for a particular microbiome or utilizing "bag-of-genes" models in which all microbial metabolic capabilities are combined into one network (34–37) (Fig. 1, row 1). While model reconstruction has traditionally been a major bottleneck, recently developed GEM reconstruction pipelines and curated resources are eliminating many of the challenges associated with throughput and model quality (32, 36, 38–40). A remaining challenge, however, is to account for model structural uncertainty introduced through gene annotation, gene-to-reaction mapping, environment specification, and biomass composition. In combination with degeneracy in FBA solutions, this accumulates to a total uncertainty associated with model predictions that is difficult to quantify (41). Ensemble modeling is a promising approach for representing this uncertainty, but challenges remain in scaling these approaches to address heterogeneous sources of uncertainty at the microbiome level (42, 43).

Scaling the scope of GEM simulations to communities (Fig. 1, row 2) results in a number of additional sources of uncertainty and technical challenges. First, different chemical and reaction namespaces between disparate data sets and models can hinder the combination of GEMs from different sources (44, 45). Additionally, community FBA methods require the formulation of a community-level objective that both draws from biological rationales and balances the trade-off between community-level and individual objectives, making it difficult to accurately predict phenotypes for complex microbiomes (21, 46).

Instead of defining a community-level objective, microbial interactions mediated via the environment can be simulated through dynamic FBA (Fig. 1, row 3), a technique that allows community dynamics to emerge from individual-level objectives. However, dynamic FBA requires additional model parameters (e.g., metabolite- and organism-specific uptake kinetics) that are challenging to estimate for a complex microbiome. Spatial dynamics such as diffusion can be combined with dynamic FBA through more advanced modeling frameworks (9, 47), but this also introduces parameters which are difficult to estimate (Fig. 1, row 4). Increasing the time scale of dynamic microbiome

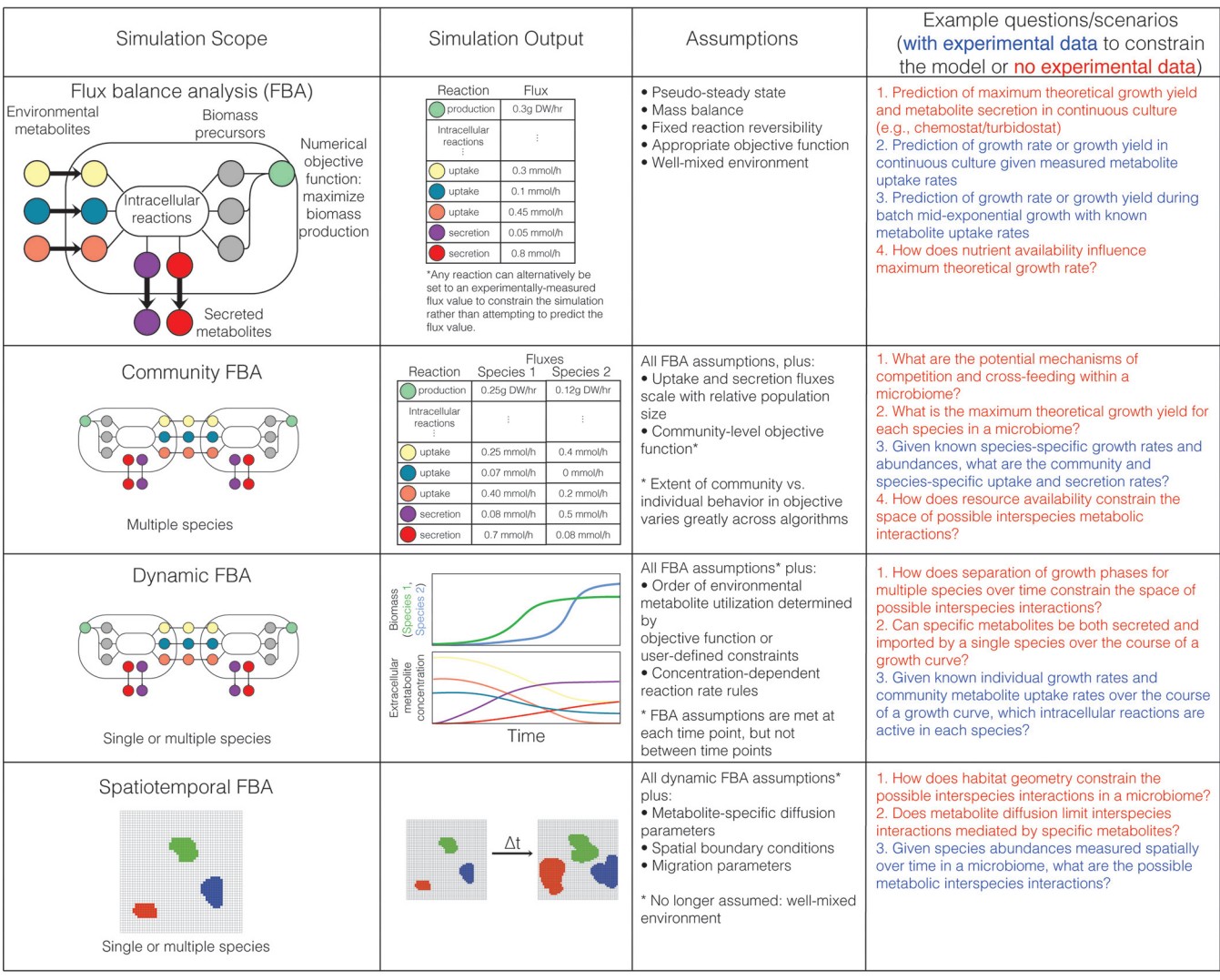

**FIG 1** Choice of microbiome modeling scope with genome-scale metabolic models (GEMs). Row 1 shows analysis of individual species (alone or within a microbiome) at a single time point with flux balance analysis (FBA). FBA can also be used for bag-of-genes models of a microbiota, which include all metabolic functions of the microbiota within a single compartment. Row 2 shows simulation of multiple species at a single time point in a community context with community FBA (where "Community FBA" is inclusive of all nondynamic FBA-based methods employing GEMs for multiple organisms in a shared extracellular compartment). Row 3 shows dynamic FBA, wherein metabolic fluxes and growth are simulated for a community with objective functions at the individual organism level. In dynamic FBA, multiple iterations of FBA are performed to introduce a temporal dimension. Community dynamics emerge in dynamic FBA through the shared extracellular compartment. Row four shows spatiotemporal FBA, which introduces spatial dimensions within a dynamic FBA framework to account for diffusion (or environmentally induced mixing) of metabolites and microbial biomass. Example simulation output and a nonexhaustive list of critical assumptions are listed for each scope. Examples of appropriate scenarios and questions are shown for each scope, with examples showing how introducing experimental data to constrain GEMs (text in blue) can increase the specificity of the simulation output at each scope relative to scenarios where little or no experimental data are used to constrain GEMs (text in red). Note that the following terms are not interchangeable: growth rate (growth per unit time), growth yield (growth per unit substrate uptake), and maximum theoretical growth yield (maximum growth per unit substrate, determined by stoichiometry and reversibility alone).

simulations is possible but complicates the underlying mechanisms as metabolic adaptation and evolution play into eco-evolutionary dynamics (48, 49).

As models scale up to represent dynamics at these higher levels, it is important to understand the propagation of the uncertainties mentioned above to ensure robust simulation frameworks. Efforts toward modeling microbiomes with individual-organism resolution need to consider the current state of accuracy and uncertainty in each constituent GEM, as well as the sensitivity of the phenotype of interest to this uncertainty. In the simplest case, we may consider an *in silico* study of two bacterial strains during steady-state growth with glucose as a sole carbon source. Here, the goal of the study would be to predict whether a coculture of the two organisms will increase

biomass yield for either individual organism. If the coculture simulation accounts for uncertainty, the researchers can base their degree of trust in the simulation on how the predictions are distributed. The researchers can also use uncertainty in monoculture simulations for any functions related to the coculture outcomes (growth yield in the fresh medium, metabolite secretion, and uptake and metabolism of putatively cross-fed metabolites) to identify functions in individual GEMs that should be curated to reduce uncertainty (and potentially improve accuracy) of the coculture simulations.

Phenotyping individual organisms in high throughput is a promising approach toward improving GEM curation; however, it is not always feasible (e.g., obligate intracellular organisms) or may be experimentally challenging to set up (e.g., organisms which have no known, chemically defined culture conditions) (36, 50). Furthermore, when studies are scaled to the complexity of most natural microbiomes, it becomes infeasible to test the performance of each individual organism in well-controlled monocultures. As we attempt to make use of increasingly complex models of microbiomes, systematic curation strategies will need to be coupled with expert intuition to establish the appropriate level of detail that is coarse-grained enough to be robust to parameter uncertainty and stochastic community dynamics (51–53) yet detailed enough to capture important processes. Researchers should keep this in mind when selecting the appropriate simulation scope for the application of GEMs to microbial communities and should try to balance the opportunities and challenges of each approach. We discuss the implications of these challenges for scientific funding priorities in the following section.

## MOVING FORWARD

The potential of GEMs to advance microbiome science is founded upon a rapidly growing body of genomic (and, increasingly, transcriptomic and metabolomic) data, which has already enabled researchers and clinicians to explore the metabolic functions of microbiomes associated with diverse ecosystems—from soil and plants (54, 55), to insects (56), to mammals (36, 57). However, these advances have largely been driven by independent analyses by theoretical and empirically focused research groups. Thus, they have resulted in a fragmentary understanding of the mechanisms and impact of interspecies interactions on microbiome function. While many opportunities exist for two-way communication between theoretical and empirical microbiome research communities, several barriers prevent this potential from being realized.

In order to better understand these barriers, we conducted a community survey of microbiome scientists (Fig. S1, Table S2). This survey revealed that, while over 70% of empirical researchers expressed an interest in using metabolic modeling, a lack of computational expertise and concern about the accuracy of predictions has prevented them from integrating models into their work. Based on these responses, we advocate for increased accessibility of modeling techniques and more transparent communication and interpretation of simulation results. To this end, we have compiled existing resources to initiate microbiome scientists—from a range of backgrounds—to use genome-scale modeling to address questions about the microbiome (Fig. 2). Importantly, we have highlighted resource gaps that must be filled to maximize the accessibility of the modeling field and facilitate cross talk between theoretical and empirical researchers. Such resources will enable microbiome scientists to determine the appropriateness of applying this modeling framework to their research questions.

In addition to the points raised by empirical researchers in our survey, participants with modeling experience echoed our concerns regarding the lack of universal approaches to verify model robustness and evaluate predictions. This challenge is especially important in microbiome science, given the potential for error propagation associated with scaling up models to the enormous diversity of organisms found in many microbiomes. We advocate for improving model trustworthiness via transparent modeling practices (construction, curation, and validation) and the integration of uncertainty in predictions. Just as community efforts have led to advances and

## Scientist's expertise:

**computational** ⟷ **biological**

### Biochemistry

**Textbook:** The processes of Life (Hunter)
**Website:** Intro to MetaCyc database
**Website:** Intro to fundamental biochemistry at Khan Academy

### Programming

**Primer:** 10 Simple Rules for biologists learning to program (Carey & Papin)
**Textbook:** Computing Skills for Biologists (Wilmes & Allesina)
**Textbook:** Practical Computing for Biologists (Dunn & Haddock)
**Still needed:** dedicated time to learn programming

### Microbiology

**Website:** Prokaryote structure and metabolism
**Textbook:** Microbe (Swanson et al.)
**Course:** Marine Biological Laboratory Microbial Diversity
**Still needed:** dedicated time to learn biology

### Representing Biology with Math

**Textbook:** A Biologist's Guide to Mathematical Modeling in Ecology and Evolution (Chapter 2, Otto and Day)
**Still needed:** graphics and interactive tutorials

## Addressing Educational Prerequisites

## Becoming a microbiome (metabolic) modeler

### Ecology

**Review:** Cooperation in microbial communities and their biotechnological applications (by Cavaliere et al.)
**Textbook:** Resource competition and community structure (Tilman)
**Textbook:** Bacterial Metabolism (Gottschalk)

### Flux Balance Analysis

**Primer:** What is flux balance analysis? (Orth et al.)
**Review:** Constraint-based models predict metabolic and associated cellular functions (Bordbar et al.)
**Textbook:** Systems Biology - Constraint-based Reconstruction and Analysis (Palsson)
**Textbook:** Optimization methods in metabolic networks (Maranas and Zomorrodi)

### Creating, Reusing, and Curating GEMs

**Building GEMs:** KBase, modelSEED, CarveMe, GapSeq
**Genomes for GEM construction:** RefSeq NCBI, EcoCyc, UHGP (human gut microbes), At-SPHERE (plants)
**Finding GEMs via databases:** BiGG, BioModels
**Curating GEMs:** A protocol for generating genome-scale metabolic reconstructions (Thiele & Palsson)
**GEMs format:** Community standards to facilitate development and address challenges in metabolic modeling (Carey et al.), understanding Systems Biology Markup Language (SBML)
**Still needed**: tutorials to communicate GEM structure and bridge intro topics to application areas, support for model validation

### Necessary Software & Tools

**Core Software:** COBRApy (Python) or COBRA Toolbox (MATLAB)
**Additional Algorithms/Modeling approaches:** CAFBA, dFBA, COMETS, Ensemble modeling (Medusa, CarveMe), data integration algorithms (e.g., GIMME)
**Visualization:** Escher, COBRA Toolbox, Systems Biology Graphical Notation (SBGN)

### Evaluating GEMs

**Adherence to standards, benchmarking:** MEMOTE
**Example of evaluating GEM performance using experimental data:** Reconstruction of the metabolic network of Pseudomonas aeruginosa to interrogate virulence factor synthesis (Bartell et al.)

### Simulations

**Introduction to common simulations:** A simplified metabolic network reconstruction to promote understanding and development of flux balance analysis tools (Rawls et al.)
**Visualizing flux:** Escher-FBA
**Cell factory design:** Caffeine, MEWpy
**Still needed:** tutorials for interpretation of simulations

*** Associated supplemental table includes full reference information and accession information for each resource

**FIG 2** Opportunities for training and establishing curricula to prepare scientists to apply genome-scale metabolic modeling in microbiome studies. Each topic rectangle lists resources we feel are beneficial for scientists exploring genome-scale metabolic modeling of microbiomes. All resources are listed with accession information in Table S1. The top half includes prerequisite topics based on researcher background; those with more computational experience (left side) or more biological experience (right side) will have different knowledge gaps. The bottom half includes resources that should be helpful regardless of background. Content in "still needed" lines was identified by our authors as having strong deficiencies in available learning material.

standardization in GEM validation (58), similar endeavors are needed to ensure transfer of knowledge across studies, both in specific scientific application areas and in the practice of GEM reconstruction. These efforts should focus on ensuring that GEM construction and curation processes are transparent and modifiable (59).

We close with specific recommendations for the microbiome modeling community and funding agencies to address the challenges and capture the opportunities we have highlighted.

- First, the modeling community should create new mechanisms for evaluating GEM construction and simulation methodology. Specifically, we recommend establishing competitive challenges with objective assessments (60, 61). As an example, the Critical Assessment of protein Structure Prediction (CASP) has been funded by the NIH for over 20 years at about $500,000 per year and has yielded a variety of useful methods for predicting protein structure.
- Second, funding agencies should establish mechanisms and proposal expectations that incentivize development, maintenance, and improvement of modeling methods and tools. As in bioinformatics tool development (62), genome-scale metabolic modeling methods are nearly exclusively developed *ad hoc* through grants focused on application rather than development of methods. We see two divergent paths to foster collaborative microbiome studies enabled by metabolic modeling:

  - Increase budget limits for proposals involving collaborating specialist groups.
  - Reduce expectations of interdisciplinarity in grant proposals. As nearly all modelers are interested in collaborating with experimentalists, providing funding stability which is not contingent on a collaboration will reduce grant writing burden and administrative complexity and encourage transparent communication of modeling results between collaborators. Experimentalists will also feel more comfortable only pursuing modeling approaches that are likely to add value to their work.

Early career researchers, who often choose to specialize to demonstrate productivity and compete for funding, will benefit most from reduced collaborative expectations at the grant proposal review stage and a culture of transparent modeling. Efforts to increase methodological transparency in the metabolic modeling field will also benefit early career researchers by lowering barriers to entry for researchers from diverse disciplines and backgrounds. The training resources enabled through transparency will increase access to modeling tools and allow researchers without access to costly experimental technologies greater opportunity to do science. These aspects are particularly relevant to researchers using GEMs within the microbiome field, which is already incredibly interdisciplinary. Ultimately, the challenges associated with applying GEMs to the microbiome field emphasize fundamental opportunities for growth of the genome-scale metabolic modeling field.

## SUPPLEMENTAL MATERIAL

Supplemental material is available online only.
**FIG S1**, TIF file, 2.8 MB.
**TABLE S1**, XLSX file, 0.01 MB.
**TABLE S2**, XLSX file, 0.02 MB.

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
