## [Reviewer comments · mSystems]

Enhancing microbiome research through genome-scale metabolic modeling

Nana Ankrah, David Bernstein, Matthew Biggs, Maureen Carey, Melinda Engevik, Beatriz García-Jiménez, Meiyappan Lakshmanan, Alan Pacheco, Snorre Sulheim, and Gregory Medlock

Corresponding Author(s): Gregory Medlock, Vedanta Biosciences

Review Timeline:

Submission Date:	May 14, 2021
Editorial Decision:	July 8, 2021
Revision Received:	September 19, 2021
Editorial Decision:	October 12, 2021
Revision Received:	November 1, 2021
Accepted:	November 19, 2021

Editor: Jose Pablo Dundore-Arias

Reviewer(s): The reviewers have opted to remain anonymous.

Transaction Report:

DOI: <https://doi.org/10.1128/mSystems.00599-21>

July 8, 2021

Dr. Gregory L. Medlock
University of Virginia
Pediatrics
Charlottesville, Virginia

Re: mSystems00599-21 (Enhancing microbiome research through trustworthy genome-scale metabolic modeling)

Dear Dr. Gregory L. Medlock:

Thank you for submitting your manuscript to mSystems. I appreciate your patience. We have completed our review and I am pleased to inform you that, in principle, we expect to accept it for publication in mSystems. However, acceptance will not be final until you have adequately addressed the reviewer comments. Considering the publication timeline for this special issue, we would appreciate if you can complete the revision and submit the "Response to Reviewers" by July 31st, 2021.

Both the reviewers and I believe this manuscript offers an interesting overview of the current state of genome-scale metabolic model research, highlighting opportunities for using GEMs to understand the dynamics of microbiomes, as well as challenges related to modeling quality, curation, and predictability. However, the current version of the manuscript needs some additional work as it does not provide sufficient background and explanation of some of the proposed ideas. This is particularly important for this issue aimed at highlighting multidisciplinary research perspectives and cross-systems topics for a diverse audience, therefore including non-expert readers.

The reviewers have provided excellent feedback to help improve your manuscript. Please pay close attention and address their comments. Please make sure that the correct supplementary materials and corresponding links are provided. Also, make sure that the writing style and flow of ideas is cohesive and consistent, specially if the editing/writing is done collectively by different co-authors.

Preparing Revision Guidelines

For complete guidelines on revision requirements for your article type, please see the journal Article Types requirement at <https://journals.asm.org/journal/mSystems/article-types>. **Submissions of a paper that does not conform to mSystems guidelines will delay acceptance of your manuscript.**

Sincerely,

Jose Pablo Dundore-Arias

Editor, mSystems

Journals Department
Reviewer comments:

Reviewer #1 (Comments for the Author):

Ankrah et al. provide interesting discussion on some opportunities and challenges associated with using genome-scale metabolic models to understand the dynamics of microbiomes. The primary take home is that currently models are often insufficiently curated, and that the uncertainty around predictions is not adequately assessed. Many interesting topics are addressed however the manuscript currently does not provide enough explanation of ideas to be approachable for someone not familiar with metabolic models. As is often the case with group efforts like this the flow of ideas is disconnected and inconsistent. Additionally, the manuscript seems incomplete at this time with reference to supplementary materials and links that do not yet exist. All that being said, the authors represent an exciting group from varying scientific backgrounds, and the idea of varying forms of uncertainty in model predictions is important. This manuscript has potential but it has a ways to go.

Specific comments:

Line 27 - It would be useful to provide a more explicit description of what a genome-scale metabolic model is for readers unfamiliar with them. You might also consider including a statement about what differentiates GEM from other modeling approaches that can be used to predict dynamics in microbial systems.

Line 28 - A GEM does not typically reconstruct temperature requirements among other things which are usually included in the fundamental niche.

Line 46 - What is meant by "outside of the context of in silico simulations"?

Figure 1 - It would be good to provide some explanation of flux balance analysis in the main text.

Line 50 - Many important challenges are identified however, many of these challenges are not explained adequately for a lay reader. Additionally, the logical connection between different challenges is often hard to follow.

Line 52 - What is meant by scope in this context?

Line 64 - I am not sure that most experimentalist would agree that their questions should be steered to match modelling approaches.

Line 73 - It would be useful to more explicitly state what generates uncertainty in the models. For example, you might consider mentioning that the network may be incomplete/incorrect, the bounds on reactions may be incorrect, there may be many equally optimal solutions, biomass composition may be incorrect or assumptions (such as optimality) may be violated. It may not be possible to get in all of these given space constraints, but some explicit examples would be useful for readers.

Line 76 - Naming differences certainly represent a challenge, but they doesn't seem related to the uncertainty that is the focus of this paragraph.

Line 84 - Would a GEM need to accurately predict all behavior in monoculture to accurately predict behavior in a community?

Line 87 - I was unclear on exactly what the standard was. Perhaps rephrase to something like "Testing behavior in monoculture may not always be possible..."

Line 91 - Most readers will not know what the Thiele and Palsson protocol is.

Line 94 - Change in-between to between.

Line 96 - Don't models in monoculture usually require bounds on exchange reactions?

Line 100 - I don't think that this explanation of objective functions will be interpretable by most lay readers.

Figure 2 - The figure has potential but is incomplete.

Line 121 - I could not find any supplementary material attached.

Reviewer #2 (Comments for the Author):

Overall, the perspective is timely and well-written. It (successfully) attempts to highlight the issue of modelling quality. Though I disagree with some of the statements and interpretations, I refrain myself from criticism on these since this is an opinion piece. Below are some comments that I hope the authors will find useful in improving the accuracy and logic of the narrative.

1. The definition of "validation" for model is not clear. Some referencing (especially to MEMOTE) seems to imply this as a validation step while at other places experimental validation seems expected. A clarity is warranted.
2. It reads towards the end as if the authors are advocating for experiment-based curation for single species models before community simulations. Yet, they also (correctly) state that this is an impossible task given the complexity of communities and diversity of microbes. Note that we are talking about trillions of species (not even thinking about strains - strong evidence exists that metabolic properties differ across strains).
3. "Curation" in GEM field comes with the caveat of limited understanding of experimental data and microbial physiology by many modelers. This is sadly also reflected here. The "high confidence" models with plenty of "curation" over years are based on laboratory data under limited conditions. Gene KO studies used in curation are limited to single KO. Even the best curated models fail dramatically to predict epistasis. And as stated above, strain differences abound. So, given this, would the authors say that any GEM models shouldn't be used at all (I know they would not, but the message can be misleading as it is presented now)?
4. Line 28: "GEM completely reconstructs an organism's fundamental niche"
Not true - it does so only from a, limited, nutrient requirement point of view. The concept of niche goes much beyond nutrient requirements. GEMs do not capture many of these - inhibitory and regulatory effects, Temp / pH effects, surface attachments; and even for metabolism - GEMs are so far no good for quantitative effects.
5. Figure 1, row 1, Assumptions: mass balance and thermodynamic constraints are not "assumptions" - they are the basis of science!
6. Yield and rate are interchangeably used in the Figure 1 (which is rather a Table). Though in FBA context, these are often directly linked, it is better to stay with one uniform terminology, especially for the sake of readers from outside the field.

Response to Reviewers

Review comment

Author reply

Reviewer #1

Ankrah et al. provide interesting discussion on some opportunities and challenges associated with using genome-scale metabolic models to understand the dynamics of microbiomes. The primary take home is that currently models are often insufficiently curated, and that the uncertainty around predictions is not adequately assessed. Many interesting topics are addressed however the manuscript currently does not provide enough explanation of ideas to be approachable for someone not familiar with metabolic models. As is often the case with group efforts like this the flow of ideas is disconnected and inconsistent. Additionally, the manuscript seems incomplete at this time with reference to supplementary materials and links that do not yet exist. All that being said, the authors represent an exciting group from varying scientific backgrounds, and the idea of varying forms of uncertainty in model predictions is important. This manuscript has potential but it has a ways to go.

We appreciate the reviewers interest in the topics we have presented, and have attempted to make complex topics more approachable while remaining as concise as possible given the space constraints for this opinion article. We have resolved inconsistencies in semantics and word choice and reorganized some sections to improve flow. Survey results have been completed and education resources have been compiled.

Specific comments:

Line 27 - It would be useful to provide a more explicit description of what a genome-scale metabolic model is for readers unfamiliar with them. You might also consider including a statement about what differentiates GEM from other modeling approaches that can be used to predict dynamics in microbial systems.

We have added a short statement at the beginning of the “Opportunities” section describing a GEM in simple terms for readers unfamiliar with these models. Given space constraints, we opted not to include comparisons to other modeling approaches because these comparisons are present in most reviews of genome-scale metabolic modeling for microbial communities .

Line 28 - A GEM does not typically reconstruct temperature requirements among other things which are usually included in the fundamental niche.

“fundamental niche” has been removed to avoid confusion.

Line 46 - What is meant by "outside of the context of in silico simulations"?

We have added a clarifying example expanding on Garza et al., in which GEMs are used to predict the traits of individual organisms and these traits are used as input data in followup analyses that do not involve GEMs. We hope that the meaning is now clearer.

Figure 1 - It would be good to provide some explanation of flux balance analysis in the main text.

We have added a brief description of FBA in the “Opportunities” section.

Line 50 - Many important challenges are identified however, many of these challenges are not explained adequately for a lay reader. Additionally, the logical connection between different challenges is often hard to follow.

We would like to thank the Reviewer for raising this concern, and we have now made several edits both to clarify the challenges and make the connections between them more clear. Specifically, we have structured it to follow figure 1 more closely by identifying key challenges that emerge in these different simulation scopes.

Line 52 - What is meant by scope in this context?

We appreciate that the Reviewer requests this clarification, as we believe the now-edited text makes our message clearer for the lay reader. We have changed our use of “scope” to refer to the simulation scope mentioned in figure 1. We have changed the text pointed out by the reviewer to refer to the appropriate “context” for GEM applications by which we mean meaningful research questions and model applications that are not outside the fundamental limitations of GEMs and associated methods.

Line 64 - I am not sure that most experimentalist would agree that their questions should be steered to match modelling approaches.

We agree with the reviewer that this sentence could be easily misinterpreted. We have now deleted this section and instead include a more detailed description of what was meant in the last paragraph of the challenges section.

Line 73 - It would be useful to more explicitly state what generates uncertainty in the models. For example, you might consider mentioning that the network may be incomplete/incorrect, the bounds on reactions may be incorrect, there may be many equally optimal solutions, biomass composition may be incorrect or assumptions (such as optimality) may be violated. It may not be possible to get in all of these given space constraints, but some explicit examples would be useful for readers.

As suggested by the Reviewer we now mention some of the most relevant sources of uncertainty. Again, due to space constraints, we now also more explicitly point the reader to a recent review that covers this topic in depth (Bernstein et al., 2021, Genome Biology).

Line 76 - Naming differences certainly represent a challenge, but they doesn't seem related to the uncertainty that is the focus of this paragraph.

We have left this sentence in this paragraph as we believe it is related to the “Challenges” of applying GEMs, but have clarified that this is not related to uncertainty, as the reviewer suggests.

Line 84 - Would a GEM need to accurately predict all behavior in monoculture to accurately predict behavior in a community?

We have attempted to clarify our stance on the curation/validation of individual GEMs for community modeling in the last paragraph of the challenges section. Here we emphasize the importance of establishing an appropriate simulation scope that captures the scientific questions of interest but is robust to uncertainty. The necessary level of accuracy of individual GEMs is thus dependent on the scientific question being addressed. We have also added a minimal example that introduces this logic within the challenges section via a co-culture of two bacterial strains.

Line 87 - I was unclear on exactly what the standard was. Perhaps rephrase to something like "Testing behavior in monoculture may not always be possible..."

We have rephrased this sentence as suggested by the Reviewer.

Line 91 - Most readers will not know what the Thiele and Palsson protocol is.

We would like to thank the Reviewer for pointing this out, and we have now removed this section.

Line 94 - Change in-between to between.

This section has been removed.

Line 96 - Don't models in monoculture usually require bounds on exchange reactions?

As the Reviewer correctly points out, bounds on exchange reactions are needed to simulate growth using GEMs both in monoculture and in cultures with multiple species. However, uptake rates can be estimated from cultivation data for monocultures, while this becomes increasingly difficult with increasing numbers of different species where metabolic by-products also increase medium complexity. We have rewritten this sentence to make the message more clear.

Line 100 - I don't think that this explanation of objective functions will be interpretable by most lay readers.

We agree with the Reviewer that the explanation of objective functions was incomplete, and we have now expanded this sentence to make the text more accessible to readers outside the COBRA community.

Figure 2 - The figure has potential but is incomplete.

We have completed the figure and welcome any additional resource recommendations from the reviewer.

Line 121 - I could not find any supplementary material attached.

The supplementary material has been updated with accession info for educational resources, survey data, and survey visualization.

Reviewer #2

Overall, the perspective is timely and well-written. It (successfully) attempts to highlight the issue of modelling quality. Though I disagree with some of the statements and interpretations, I refrain myself from criticism on these since this is an opinion piece. Below are some comments that I hope the authors will find useful in improving the accuracy and logic of the narrative.

1. The definition of "validation" for model is not clear. Some referencing (especially to MEMOTE) seems to imply this as a validation step while at other places experimental validation seems expected. A clarity is warranted.

We appreciate the reviewer pointing out this potentially confusing lack of clarity. We have now made the challenges section more concise, removing most references to model validation. We have briefly addressed this topic in the last paragraph of the challenges section where we hope that the terminology is more clear and consistent.

2. It reads towards the end as if the authors are advocating for experiment-based curation for single species models before community simulations. Yet, they also (correctly) state that this is an impossible task given the complexity of communities and diversity of microbes. Note that we are talking about trillions of species (not even thinking about strains - strong evidence exists that metabolic properties differ across strains).

We appreciate the point that the reviewer raises here, which points towards one of the major challenges of applying GEM analysis to microbial communities. We have attempted to clarify our stance in the last paragraph of the challenges section. Here we state that we believe the appropriate path forward lies in selecting a level of detail that captures important processes while remaining robust to uncertainty. We hope that our perspective will provide some additional tools for researchers to approach this difficult question.

3. "Curation" in GEM field comes with the caveat of limited understanding of experimental data and microbial physiology by many modelers. This is sadly also reflected here. The "high confidence" models with plenty of "curation" over years are based on laboratory data under limited conditions. Gene KO studies used in curation are limited to single KO. Even the best curated models fail dramatically to predict epistasis. And as stated above, strain differences abound. So, given this, would the authors say that any GEM models shouldn't be used at all (I know they would not, but the message can be misleading as it is presented now)?

We agree with the reviewers sentiment that there are major limitations to the utility of existing GEMs (even highly curated ones). The lack of communication of these limitations in the published literature is one of the primary points of concern that we raise, and our surveyed researchers raised.

As stated in response to comment #2, the clarification of the challenges section should guide readers away from thinking we are overly cynical about the applicability of GEMs.

4. Line 28: "GEM completely reconstructs an organism's fundamental niche"

Not true - it does so only from a, limited, nutrient requirement point of view. The concept of niche goes much beyond nutrient requirements. GEMs do not capture many of these - inhibitory and regulatory effects, Temp / pH effects, surface attachments; and even for metabolism - GEMs are so far no good for quantitative effects.

As the Reviewer correctly points out there are many aspects of an organism's fundamental niche that are not covered by GEMs in their current realization. We have rewritten this paragraph and removed this sentence.

5. Figure 1, row 1, Assumptions: mass balance and thermodynamic constraints are not "assumptions" - they are the basis of science!

We disagree with the reviewer but also feel that the disagreement is mostly semantic--any phenomenon included in a model constitutes an assumption, because all models represent a limited band of mechanistic resolution. For example, mass balance is not universal--it is an assumption of the modeling framework. We can illustrate this with photosynthesis: photon mass is converted to bond energy as described by mass-energy equivalence. If the entire modeling framework accounts for mass-energy equivalence, this phenomenon can be represented exactly, but this is not the convention, and representation of photosynthesis in GEMs is widely considered non-standard. Thus mass balance is an assumption of the modeling framework, and we feel it is important to communicate because the stoichiometric foundation of GEMs is key to understanding them. With regard to "thermodynamics", we have made our intent more explicit by changing it to "Fixed reaction reversibility" to indicate that reversibility of each reaction is explicit and it does not change in the standard FBA modeling context.

6. Yield and rate are interchangeably used in the Figure 1 (which is rather a Table). Though in FBA context, these are often directly linked, it is better to stay with one uniform terminology, especially for the sake of readers from outside the field.

While we appreciate the reviewer's careful eye, yield and rate are used very precisely within the figure. The examples for the FBA scope are intended to communicate this: if uptake rates are known (example 2 & 3), FBA can be used to predict growth rates. Otherwise, if uptake rates are not known (example 1), FBA predictions are only interpreted as maximum theoretical yields (i.e., the simulated growth rate scales with unbounded uptake rates, thus the rate is arbitrary but the yield has meaning). Uses of "growth yield" and "growth rate" are similarly specific for the examples in the other modeling scopes. To help readers understand that these terms are not being used synonymously, we have made the following changes:

- 1. For FBA examples 2 & 3, changed "growth rate" to "growth rate or growth yield"; simulation of growth rates and yields are both possible with uptake rate data, whereas only the maximum theoretical growth yield can be calculated without uptake rate data.**
- 2. We have added a note to the figure caption explaining the difference between growth rate, growth yield, and maximum theoretical growth yield; we have also noted in the caption that the terms are not used interchangeably in the figure.**

October 11, 2021

Dr. Gregory L. Medlock
Vedanta Biosciences
Cambridge, Massachusetts

Re: mSystems00599-21R1 (Enhancing microbiome research through trustworthy genome-scale metabolic modeling)

Dear Dr. Gregory L. Medlock:

Thank you for submitting your manuscript to mSystems. We appreciate your patience and hard work on this perspective, and are quite excited to have this piece in our collection. Upon review of the revised version, we have further comments to be addressed before final acceptance of this perspective for publication in mSystems. However, acceptance will not be final until you have adequately addressed the recommended changes. We would greatly appreciate if you can complete these final revisions and submit the "Response to Reviewers" by October 31st, 2021.

Overall, as noted, both the reviewers and I believe this will be a great addition to our collection. This piece occupies a unique position in the special edition, being the only one to touch on the current state of genome-scale metabolic model research, and to highlight opportunities, challenges, and future directions for using GEMs to understand and model microbiome dynamics. However, the revised version is nearly twice as long as the previous submission, and well over the requested word limit (1500-1700 words), with 3102 words in the body of the manuscript alone. While I recognize that the first round of reviews suggested additional background and explanation to help the reader, we ask that you and your co-authors make one more hard edit to cut the body of the manuscript to 2200 words maximum to fit within the overall collection (most of which are within the 1500-word limit). We recognize the complexity and importance of this topic, and want to do what we can to support this piece. As previously noted, the goal of this special issue is to highlight multidisciplinary research perspectives and cross-systems topics for a diverse audience, with a special focus on early-career scientists. Thus, in editing your piece, we ask that you work to avoid overly technical terms and jargon, and work to provide an introduction to the topic that motivates novice readers to think more broadly about potential applications of GEMS to their own field. Practically, the reviewers also recommend reducing the length of some extremely-long sentences and paragraphs (e.g. paragraphs 96-111, 174-220, 225-208, 226-243), as well as non-essential sentences (lines 68-70)

Further specific comments from the reviewers are summarized below:

Title: Please eliminate the word "trustworthy"

Opportunities: Sentences within lines 28-32 are overly complex and not well explained, and should be eliminated. Sentence 56-59, mention briefly what type of statistical analysis are being referred.

Challenges: Many of the challenges discussed in this section are captured in Fig 1. Please edit to avoid overlap and duplication of information by providing concise points and referring to Fig1 for more detail. Sentences 63-74 need to be shortened by merging redundant sentences and creating new ones that provide a clear and simple explanation of the contrasting points presented. Sentences 75-95 and 126-135 highlight current technological challenges, which are likely to be overcome in the future. These sentences need to be shortened, highlighting the main points more broadly, distinguishing existing and foreseeable limitations. Bullet points may be useful for conveying key information points while cutting length. Sentences 126-150 need to be simplified as they provide overly detailed, and redundant information. Rather than referring to the co-culture scenario multiple times, this example should be used to clearly illustrate (simultaneously) the main challenges in data curation and simulation.

Moving forward: Sentences 153-161 raise concerns about broader challenges that fall outside the scope of this perspective, and therefore, should only be introduced briefly and avoid redundancy with the description of the survey results. Sentences 162-200, rather than providing detailed explanation of participant answers, highlight the take home messages of the survey results, what came across as the main community priorities, gaps and needs, and how they influence the advancement of microbiome research. The conclusion paragraphs (sentences 208-243) need to be shortened. While we recognize the need to advocate for more funding, this should be done more concisely avoiding specific examples and how lack of funding prevents us from moving forward. Instead, this would be a good opportunity to provide a more optimistic perspective, highlighting how new/additional funding can help us address remaining gaps and needs in technical capacities, as well training of the next generation of microbiome scientists, as these areas represent equally challenges and opportunities and have significant potential to stimulate discussion and motivate change within the microbiome research community.

Thank you again for your continuing efforts on this piece. We are hopeful that these edits won't be overly complicated for your team, and I am happy to communicate further as needed. Our goal is to get the final pieces of the collection over the finish line this month, with vigorous publicity for the special edition in November. Please don't hesitate to contact me if you have any

questions on the recommended edits.

Below you will find instructions from the mSystems editorial office and comments generated during the review.

Preparing Revision Guidelines

Sincerely,

Jose Pablo Dundore-Arias

Editor, mSystems

Journals Department
Reviewer comments:

Response to reviews (Responses in bold)

Overall, as noted, both the reviewers and I believe this will be a great addition to our collection. This piece occupies a unique position in the special edition, being the only one to touch on the current state of genome-scale metabolic model research, and to highlight opportunities, challenges, and future directions for using GEMs to understand and model microbiome dynamics. However, the revised version is nearly twice as long as the previous submission, and well over the requested word limit (1500-1700 words), with 3102 words in the body of the manuscript alone. While I recognize that the first round of reviews suggested additional background and explanation to help the reader, we ask that you and your co-authors make one more hard edit to cut the body of the manuscript to 2200 words maximum to fit within the overall collection (most of which are within the 1500-word limit). We recognize the complexity and importance of this topic, and want to do what we can to support this piece. As previously noted, the goal of this special issue is to highlight multidisciplinary research perspectives and cross-systems topics for a diverse audience, with a special focus on early-career scientists. Thus, in editing your piece, we ask that you work to avoid overly technical terms and jargon, and work to provide an introduction to the topic that motivates novice readers to think more broadly about potential applications of GEMS to their own field. Practically, the reviewers also recommend reducing the length of some extremely-long sentences and paragraphs (e.g. paragraphs 96-111, 174-220, 225-208, 226-243), as well as non-essential sentences (lines 68-70)

We have made major cuts to the manuscript and improved conciseness and are now at approximately 2,000 words in the main text.

Further specific comments from the reviewers are summarized below:
Title: Please eliminate the word "trustworthy"

Done.

Opportunities: Sentences within lines 28-32 are overly complex and not well explained, and should be eliminated. Sentence 56-59, mention briefly what type of statistical analysis are being referred.

Done.

Challenges: Many of the challenges discussed in this section are captured in Fig 1. Please edit to avoid overlap and duplication of information by providing concise points and referring to Fig1 for more detail. Sentences 63-74 need to be shortened by merging redundant sentences and creating new ones that provide a clear and simple explanation of the contrasting points presented. Sentences 75-95 and 126-135 highlight current technological challenges, which are likely to be overcome in the future. These sentences need to be shortened, highlighting the main points more broadly, distinguishing existing and foreseeable limitations. Bullet points may be useful for conveying key information points while cutting length. Sentences 126-150 need to be simplified as they provide overly detailed, and redundant information. Rather than referring to the co-culture scenario multiple times, this example should be used to clearly illustrate (simultaneously) the main challenges in data curation and simulation.

We have added additional references in the text to Figure 1 and cut adjacent text. The entire section has been shortened substantially, including the specific sentences requested.

Moving forward: Sentences 153-161 raise concerns about broader challenges that fall outside the scope of this perspective, and therefore, should only be introduced briefly and avoid redundancy with the description of the survey results. Sentences 162-200, rather than providing detailed explanation of participant answers, highlight the take home messages of the survey results, what came across as the main community priorities, gaps and needs, and how they influence the advancement of microbiome research. The conclusion paragraphs (sentences 208-243) need to be shortened. While we recognize the need to advocate for more funding, this should be done more concisely avoiding specific examples and

how lack of funding prevents us from moving forward. Instead, this would a good opportunity to provide a more optimistic perspective, highlighting how new/additional funding can help us address remaining gaps and needs in technical capacities, as well training of the next generation of microbiome scientists, as these areas represent equally challenges and opportunities and have significant potential to stimulate discussion and motivate change within the microbiome research community.

We have removed redundancies and greatly simplified this section.

November 19, 2021

Dr. Gregory L. Medlock
Vedanta Biosciences
Cambridge, Massachusetts

Re: mSystems00599-21R2 (Enhancing microbiome research through genome-scale metabolic modeling)

Dear Dr. Gregory L. Medlock:

Your manuscript has been accepted, and I am forwarding it to the ASM Journals Department for publication. For your reference, ASM Journals' address is given below. Before it can be scheduled for publication, your manuscript will be checked by the mSystems senior production editor, Ellie Ghatineh, to make sure that all elements meet the technical requirements for publication. She will contact you if anything needs to be revised before copyediting and production can begin. Otherwise, you will be notified when your proofs are ready to be viewed.

Since this perspective is part of the Special Series Deciphering the Microbiome, we would like to broadly publicize this perspective. Once the final version of the manuscript is published, we ask that you and/or your coauthors share the link on social media to amplify its visibility. Other pieces published earlier have done this and got substantial attention. If you share on Twitter, we ask you to please Thank NSF using the #NSFMicrobiomes, as well as the editorial team: @lindakinkel, @klassenlab, @lupolabs, @ashley17061, @napaaqtuk and @jp_dundorearias.

Publication Fees: THIS DOES NOT APPLY TO YOU AS THIS IS PART OF AN SPONSORED SPECIAL ISSUE

We recognize that the video files can become quite large, and so to avoid quality loss ASM suggests sending the video file via <https://www.wetransfer.com/>. When you have a final version of the video and the still ready to share, please send it to Ellie Ghatineh at eghatineh@asmusa.org.

Sincerely,

Jose Pablo Dundore-Arias
Editor, mSystems

Journals Department
Table S1: Accept
Figure S1: Accept
Table S2: Accept